# Enhanced Multi-Strategy Particle Swarm Optimization for Constrained Problems with an Evolutionary-Strategies-Based Unfeasible Local Search Operator

**Marco Martino Rosso** [1] , **Raffaele Cucuzza** [1,*] , **Angelo Aloisio** [2] **and Giuseppe Carlo Marano** [1]

1   DISEG, Department of Structural, Geotechnical and Building Engineering, Politecnico di Torino,
    Corso Duca degli Abruzzi, 24, 10128 Turin, Italy; marco.rosso@polito.it (M.M.R.);
    giuseppe.marano@polito.it (G.C.M.)
2   Civil Environmental and Architectural Engineering Department, Università degli Studi dell'Aquila,
    Via Giovanni Gronchi n.18, 67100 L'Aquila, Italy; angelo.aloisio1@univaq.it
*   Correspondence: raffaele.cucuzza@polito.it

**Abstract:** Nowadays, optimization problems are solved through meta-heuristic algorithms based on stochastic search approaches borrowed from mimicking natural phenomena. Notwithstanding their successful capability to handle complex problems, the No-Free Lunch Theorem by Wolpert and Macready (1997) states that there is no ideal algorithm to deal with any kind of problem. This issue arises because of the nature of these algorithms that are not properly mathematics-based, and the convergence is not ensured. In the present study, a variant of the well-known swarm-based algorithm, the Particle Swarm Optimization (PSO), is developed to solve constrained problems with a different approach to the classical penalty function technique. State-of-art improvements and suggestions are also adopted in the current implementation (inertia weight, neighbourhood). Furthermore, a new local search operator has been implemented to help localize the feasible region in challenging optimization problems. This operator is based on hybridization with another milestone meta-heuristic algorithm, the Evolutionary Strategy (ES). The self-adaptive variant has been adopted because of its advantage of not requiring any other arbitrary parameter to be tuned. This approach automatically determines the parameters' values that govern the Evolutionary Strategy simultaneously during the optimization process. This enhanced multi-strategy PSO is eventually tested on some benchmark constrained numerical problems from the literature. The obtained results are compared in terms of the optimal solutions with two other PSO implementations, which rely on a classic penalty function approach as a constraint-handling method.

**Keywords:** particle swarm optimization (PSO); multi-strategy PSO; self-adaptive evolutionary strategies (ES); local search operator; constraints handling

---

## 1. Introduction

In optimization problems, the aim is optimizing certain mathematical functions, called Objective Functions (OF) $f(\boldsymbol{x})$. These problems can be divided into single-objective or multi-objective problems, depending on the number of OFs, and a further subdivision for single-objective problems is based on the presence of constraints. Unconstrained problems are defined as:

$$\min_{\boldsymbol{x}\in\Omega}\{f(\boldsymbol{x})\} \tag{1}$$

meanwhile, constrained problems are defined as:

$$
\begin{aligned}
\min_{\boldsymbol{x}\in\Omega}&\{f(\boldsymbol{x})\}\\
\text{s.t.}\quad & g_q(\boldsymbol{x})\leq 0 \quad \forall q=1,\ldots,n_q\\
& h_r(\boldsymbol{x})=0 \quad \forall r=1,\ldots,n_r
\end{aligned}
\tag{2}
$$

where $x = \{x_1, \ldots, x_j, \ldots, x_n\}^T$ is the design vector whose terms are the parameters to be optimized. The search domain is a multidimensional space $\Omega$ based on the admissible intervals of values for each $j$-th variable, which are defined by its lower and upper bounds $[x_j^l, x_j^u]$. This detects a box-type hyper-rectangular search space $\Omega$, which is typically defined as the Cartesian product (denoted by the $\times$ symbol) among the admissible intervals:

$$\Omega = [x_1^l, x_1^u] \times \ldots \times [x_j^l, x_j^u] \times \ldots \times [x_n^l, x_n^u] \tag{3}$$

The constraints in (2) can belong to two different categories: inequality $g_q(x)$ and/or equality $h_r(x)$ constraints. Each equality constraint can be easily converted into a couple of inequality constraints; therefore, without any loss of generality, it is possible to consider only inequality constraints in (2), i.e., $g_p(x) \leq 0$, where $p = 1, \ldots, n_q, n_{q+1}, \ldots, n_p$, being $n_p = n_q + 2n_r$.

The adoption of evolutionary algorithms (EAs) has received much more attention in recent years because of their successful capability to handle complex optimization problems. This is addressed mainly to the fact that they do not require any first-order (gradient) or second-order (Hessian) information coming from the problem to be solved, which is conversely a prerogative of the traditional gradient-based mathematical search approaches. Furthermore, the quite simple implementation of EAs has determined their rapid spread, and they have immediately become an attractive tool among practitioners. Among the many alternatives available nowadays, the genetic algorithm (GA) proposed by J. Holland in the 1970s [1] still represents one of the most popular population-based tools, which tries to simulate the biological evolutionary process of a set of candidates solutions mimicking the biological Darwinian Theory. This is realized by adopting specific pseudo-random-based operators such as crossover, mutation, and selection in order to reproduce the long-term process of evolution in a population with the survival of the fittest individuals [2]. In the last two decades, the adoption of metaheuristic algorithms in many engineering applications highlighted their successful capabilities to deal with real-world constrained problems [3–8], e.g., dealing with structural design [9–12] and structural optimization tasks [13–16].

In the framework of EAs, a more recent but already well-known approach is the particle swarm optimization (PSO) algorithm. It was mentioned by Kennedy and Eberhart [17] in 1995 for the first time, and then it rapidly became widespread during the following years. Contributions from the Scientific Community have not ended yet, and still nowadays there is active research about this topic to improve the search operators and the performances. The PSO is also a population-based algorithm which takes inspiration from the study of the behavioural models of birds flocking or fish schooling, whose individuals explore the natural environment in order to find and reach some source of food. Similarly, the algorithm tries to evolve a particle swarm of candidate solutions in the domain search space in order to find the optimum. The PSO was originally developed to face unconstrained problems, but it was later adapted to also solve constrained problems exploiting specific strategies.

The following section presents a brief review of the PSO mechanisms, and the main adopted strategies to solve constrained problems are mentioned. After that, the description of the proposed enhanced multi-strategy PSO method is illustrated. Finally, the authors try to merge several state-of-the-art concepts to obtain an improved PSO algorithm to successfully handle constrained problems with a non-penalty based approach. The novel contributions of this article can be summarized as follows:

- PSO implementation with the main state-of-art improvements, adopting a multi-strategy approach. In this way, the algorithm attempts to avoid wasting many iterations when the algorithm stalls or is trapped in local minima, etc.;
- A non-penalty approach for constraint handling which instead exploits information of swarm positions in terms of the objective function and the actual degree of constraint violation to guide the swarm evolution;

- A novel unfeasible local search operator is presented to help the PSO when it stalls in an unfeasible region quite close to the actual feasible one. This local search operator relies on the meta-heuristic, self-adaptive Evolutionary Strategy (ES) approach, which does not require any other further arbitrary parameter.

In a different recent contribution of the authors [18], some further novel approaches to deal with constraints have been presented, considering a hybridization of the PSO with a machine learning support vector machine. However, the current paper presents a completely different approach based on handling constraints directly based on information which can be retrieved from the swarm positions in terms of objective function and constraints violations. Finally, the enhanced multi-strategy PSO is successfully tested on some benchmark constrained mathematical problems from the literature compared with other PSO implementations that adopt more standard penalty-based constraint handling techniques. In conclusion, the proposed multi-strategy PSO has been validated on real-world case studies, considering some literature on three-dimensional truss design structural optimization problems.

## 2. Review of PSO and Constraint Handling Approaches

The PSO algorithm was directly inspired by biological behavioral models of birds flockings, school fishing or swarming of insects. In nature, these animals adopt a collective behaviour to ensure their survival, even though each individual acts as an intelligent independent entity making its own decisions. Mimicking this trend, Kennedy and Eberhart in 1995 proposed a first model of the PSO algorithm [17]. The PSO algorithm encodes a population of candidate solutions in the search space, which is composed of a certain number $N$ of intelligent agents. Although the latter can independently move inside the domain, in order to ensure an emerging intelligent collective behaviour toward the optimum, the dynamic movement of each agent is affected by some information obtained from the swarm. One of the first proposed methods is related to a Newtonian dynamics perspective, in which each $i$-th particle (with $i = 1, \ldots, N$, where $N$ is the population size) is completely defined by its position $^k x_i$ and its velocity $^k \mathbf{v}_i$ at the $k$-th generation. The velocity is thus updated taking into account two main kinds of information: First, the self-cognitive memory of each particle, which is related to the so far best visited position $^k x_i^{Pb}$ (cognitive term) and, second, the attraction toward the other particles' best visited positions $^k x^{Gb}$ (social term). Therefore, the position and the velocity of the $i$-th particle in the next $k + 1$ iteration can be written as:

$$^{(k+1)}\mathbf{v}_i = {}^k\mathbf{v}_i + c_1\,{}^{(k+1)}r_{1i} * \left[{}^k x_i^{Pb} - {}^k x_i\right] + c_2\,{}^{(k+1)}r_{2i} * \left[{}^k x_i^{Gb} - {}^k x_i\right], \tag{4}$$

$$^{(k+1)}x_i = {}^k x_i + \tau\,{}^{(k+1)}\mathbf{v}_i \quad (\tau = 1), \tag{5}$$

where the symbol $*$ denotes the term-by-term vector multiplication (Hadamard product, [19]), and the positive scalar acceleration factors $c_1$ and $c_2$ are denoted as *cognitive* and *social* parameters, respectively. The terms $^{(k+1)}r_{1i}$, $^{(k+1)}r_{2i} = rand[0, 1]$ are two random weights of the social and cognitive terms, respectively. These terms are fundamental for the purpose to introduce some randomized behaviour inside this quite deterministic model with the aim of enhancing the exploration capabilities of the model. The cognitive term is also denoted as *pbest*, whereas the social term is denoted as *gbest* when it is referred to the best global visited position so far among all the particles of the swarm. This explains why this latter strategy is also known as the *gbest PSO* model [20]. Later studies revealed that a good practise is to protect the cohesion of the swarm by restricting the velocity component to a maximum value, typically assumed as $\mathbf{v}^{\max} = \gamma(x^u - x^l)/\tau$, where $\tau = 1$ is a time-related parameter, whereas $\gamma \in [0.1, 1]$ (generally set to 0.5) defines how far a particle can move starting from its current position [21]. The typical stopping criterion of the PSO is generally set as a maximum number of iterations $k_{max}$. However, a predetermined maximum number of iterations for each problem is not usually known in advance, therefore, one can refer

to the suggestions of [22] or conduct experimental trial and error tuning of the minimum $k_{max}$, which allows one to achieve the optimum, reducing the overall computational cost. Later on, for the sake of improving the exploration capacity of the swarm, [23] introduced an inertia weight term $^{k}w$ multiplied to the current $k$th velocity in the update rule (4). This parameter can be a constant or a variable with respect to the iterations flow, e.g., from an initial value $^{0}w$ to a final one $^{L}w$ with a linearly decreasing law, but there are also many other variants in [20]. The performance of the algorithm is strongly affected by the choice of the parameters such as the swarm size $N$, usually set in a range of $[20, 100]$ with $n \leq 30$, or such as the acceleration factors, which are usually assumed statically fixed to $c_1 = c_2 = 2$ [21]. In this study, it is assumed which of all of them are constant values equal to $c_1 = c_2 = 2$, $^{0}w = 0.90$ and $^{L}w = 0.40$ [24].

One of the most important aspects to enhance the PSO performances is to improve the way in which the information are exchanged among the particles. With efficient information sharing, the swarm can exhibit a better collective convergent behaviour. The information exchange is related to the structure of the neighbourhood of each particle, which is denoted as *neighbourhood topology*. This kind of implementation is also called a local PSO model or simply *lbest model* to differentiate it from the classical so-called global PSO model or simply *gbest model* [1,20,21]. The classical gbest model approach can also be regarded as a neighbourhood strategy in which the neighbourhood is composed of the entire population. In this sense, the swarm is denoted as fully informed or fully connected. A schematic graphical representation of the swarm with the information flows is depicted in Figure 1a. The main negative aspect of this latter strategy is the greater inclination to premature convergence. If the global attractor gbest is entrapped in a local minimum, the entire swarm may probably fall down in the same local minimum without a sufficient exploration capability. The enhancement of the PSO was performed by a counter-intuitive approach which relies on slowing down the rapid convergence attitude of the PSO through channelling and limiting the information exchange, the neighbourhood concept indeed [1,20,21]. In the lbest models, it is necessary to define, firstly, the structure of the neighbourhood which controls the way in which the particles are interconnected and, secondly, the size of the neighbourhood which affects the influence of the swarm on each particle [1]. Considering the most popular time-invariant neighbourhood topologies, the ring topology is one of the easiest to be implemented, and it has also been adopted in the present study. As illustrated in Figure 1b,c, each particle in this topology forms a neighbourhood considering the nearest particles (nearest indices in a vector of positions), resulting in an ideal circular interconnection. The total number of the particles which belongs to the neighbourhood is denoted as radius $R$, as depicted in Figure 1b, in which $R = 2$, and (c), in which $R = 4$. These methods can be implemented considering that each particle in the numerical vector has a unique index, therefore, each particle can unequivocally be selected to enter in a neighbourhood through its index [25], as schematically depicted in Figure 2. A very great number of different neighbourhood topologies were developed in the last decades as showed in [25,26]. Some other implementations also involve a dynamic update of the neighbourhood size, which identifies new types of lbest models which are denoted as multi-populations or multi-swarm PSO, such as in [27].

*State of the Art of Constraint Handling*

In order to adapt EAs to deal with constrained problems, several strategies were developed by the scientific community. As a matter of fact, constraint handling is a big challenge because it is related to find the optimal point respecting all the constraints, and therefore, the algorithms may be able to deal with unfeasible solutions in an efficient way. Despite several studies (e.g., [28]) demonstrating that PSO has a good convergence rate, it was originally proposed to solve unconstrained optimization problems, such as many other Soft Computing techniques. The implementation of some effective constraint-handling mechanisms is a crucial issue for all biologically inspired optimizers [29–32]. The several

strategies developed have been classified by different authors into basically five main categories (see, for instance, the state-of-the-art review by [30,33,34]):

- Penalty-functions-based methods;
- Methods based on special operators and representations;
- Methods based on repair algorithms;
- Methods based on the separation between OFs and constraints;
- Hybrid methods.

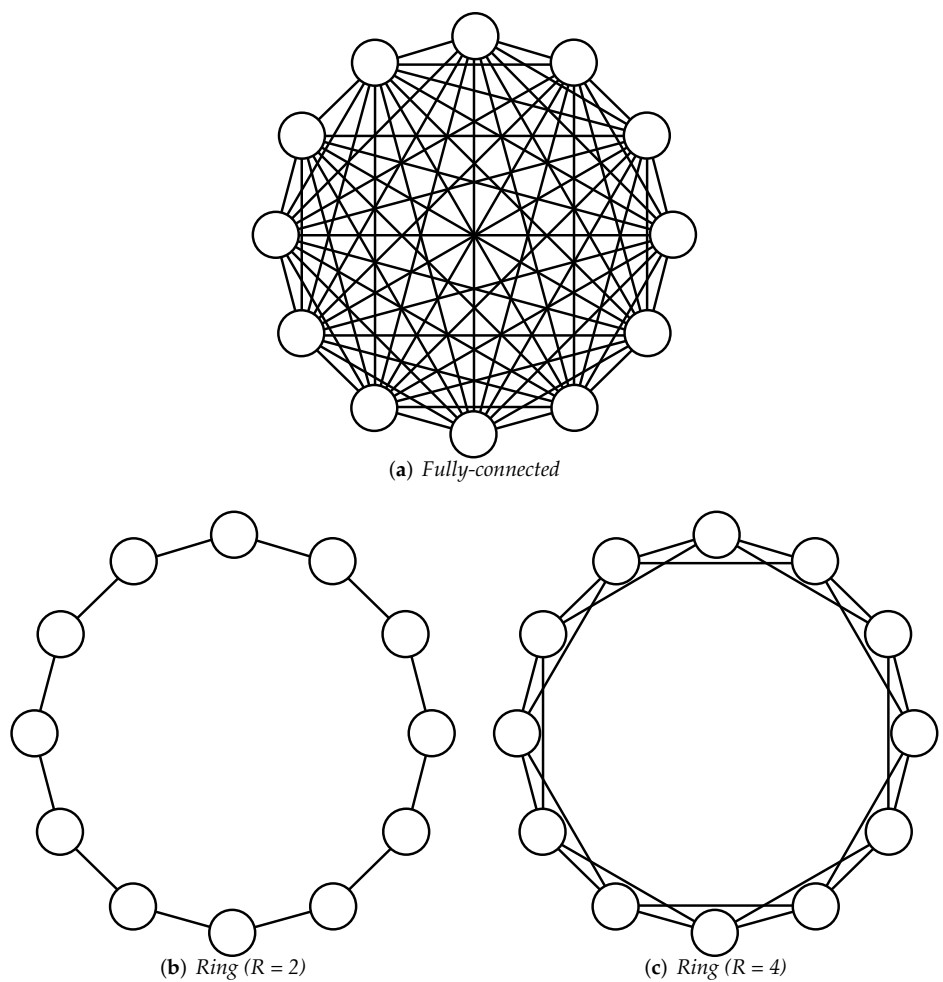

(**a**) *Fully-connected*

(**b**) *Ring (R = 2)*          (**c**) *Ring (R = 4)*

**Figure 1.** Some examples of PSO Neighborhood Topologies.

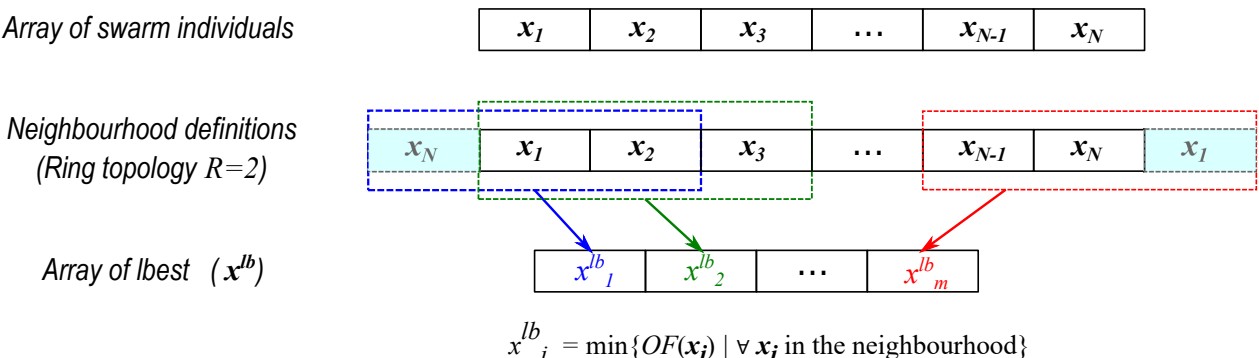

**Figure 2.** Graphical schematization of the Ring topology implementation (*R* = 2).

The most adopted method due to its simplicity is the exterior penalty approach which allows to convert the problem in an unconstrained version [35,36]. Many different approaches such as the death, static, dynamic, or adaptive penalty functions have been proposed in time, e.g., one can refer to [35]. A proper choice of the constraint-handling mechanism affects the performance of the algorithm, and one of the critical issues to take into account is the preservation of the diversity of the population. The brutal elimination of the unfeasible particles, such as in the death penalty rule, can jeopardize the exploration performances due to a loss of information [30,37]. In general, the penalty approach rely on the evaluation of a factor that applies a certain penalty to the OF, depending on the degree of violation and the number of violated constraints. Therefore, the constrained OF $f(x)$ is transformed into an analogous unconstrained OF $\phi(x)$:

$$\min_{x \in \Omega}\{\phi(x))\} = \min_{x \in \Omega}\{f(x) + H(x)\} \tag{6}$$

where $H(x)$ is the penalty function, whose specific definition depends on the strategy adopted. If the penalty is constant during the iterations, it is a *static penalty function*, while if it is changing at each iteration, it is addressed as a *dynamic penalty function*. These two techniques are the most popular tools in structural optimization, see, for instance, the papers by Hasançebi et al. [38] and Dimopoulos [39].

In the case of static-penalty-based techniques, the equivalent unconstrained problem is formulated with a static penalty factor $H_s(x)$ that is generally expressed as follows (see [40,41]):

$$H_s(x) = w_1 H_{NVC}(x) + w_2 H_{SVC}(x) \tag{7}$$

where $H_{NVC}$ is the number of constraints that are violated by the particle $x$, $H_{SVC}$ is the sum of all violated constraints, and $w1$ and $w2$ are static control parameters of the penalty scheme:

$$H_{SVC}(x) = \sum_{p=1}^{n_p} \max\{0, g_p(x)\} \tag{8}$$

The numerical values adopted by Parsopoulos and Vrahatis [40] are $w_1 = w_2 = 100$. In the present research, some standard penalty PSO approaches are adopted for making comparisons with the enhanced PSO version, which is presented in the following section. For these PSOs with penalty approaches, $w_1 = 0$ and $1000 < w_2 < 10,000$ have been assumed, depending on the analysed problem. Depending on the values of $w_1$ and $w_2$, it is possible to set the level of severity of the constraint violations: In case of extremely high control parameters, the penalty is called the *death penalty*, and it tries to completely avoid any kind of research inside the unfeasible region, even if the number of violated constraints is rather limited.

The popularity of the penalty function technique is due to its simple implementation, and it strongly enhances the performance of an algorithm that is trying to solve constrained optimization problems. To improve the effectiveness of the penalty factor, a penalty function which changes the weight of the penalty during the iterations is also adopted in the current study. Indeed, it is possible to better control the search space of the particles with this latter dynamic approach, allowing a more relaxed constraint handling at the beginning and an increasing penalty value approaching the end of the available iterations. Firstly proposed by Parsopoulos and Vrahatis [42], it has recently been adopted by Barakat and Altoubat [43] for the optimum design of RC water tanks. To this end, the (7) is readily modified as follows:

$$\min_{x \in \Omega}\{f(x) + {}^k h H_d(x)\} \tag{9}$$

in which ${}^k h$ is a dynamic penalty whose numerical value was evaluated as ([42,43]):

$$^k h = \sqrt{k} \tag{10}$$

and $H_d(x)$ is the dynamic penalty factor:

$$H_d(x) = \sum_{p=1}^{n_p} \theta_p(x)[\max\{0, g_p(x)\}]^{\gamma_p(x)} \tag{11}$$

Typical assignments for the penalty parameters are (see, for instance, [42,43]):

$$\theta_p(x) = \begin{cases} 10 & \text{if } \max\{0, g_p(x)\} \le 0.001 \\ 20 & \text{if } 0.001 < \max\{0, g_p(x)\} \le 0.100 \\ 100 & \text{if } 0.100 < \max\{0, g_p(x)\} \le 1.000 \\ 300 & \text{otherwise.} \end{cases} \tag{12}$$

$$\gamma_p(x) = \begin{cases} 1 & \text{if } \max\{0, g_p(x)\} \le 1 \\ 2 & \text{otherwise.} \end{cases} \tag{13}$$

It is evident that dynamic penalty methods require a larger number of control parameters in comparison to the static one. Considering $^k h$ as defined in (10), in the present paper, the dynamic penalty factor is assumed to have:

$$10 < H_d(x) < 1000 \tag{14}$$

The evaluation of a proper penalty is a fundamental passage to achieve a good solution of an optimization problem: Ideally, it should be set as low as possible to avoid high computational efforts and problems arising when the global optimum is close to the constraint. Indeed, if the optimum is at the boundary and the penalty is too high, the element which is attracted by that area is immediately pushed back when the boarder is crossed. This mechanism is avoided by adopting a low penalty that is not too severe in case of small violations and also allows a good investigation in such critical areas. However, if the penalty is too low and it does not contrast the constraint violation properly, a lot of effort will be spent in the unfeasible region, providing no useful information for the minimization purpose.

## 3. Enhanced PSO with a Multi-Strategy Implementation and Hybridisation with an ES-Based Operator

In the present work, starting from the standard Newtonian-dynamics-based PSO approach proposed by Kennedy and Eberhart (1995) in [17], an enhanced PSO is implemented adopting some of the most well-known available strategies in literature and adding a special operator in order to increase the search performance of the standard version. The various strategies are merged together, and the flowchart of the implemented algorithm is illustrated in Figure 3.

At first, the initial population is generated randomly in the hyper-rectangle search space, adopting the Latin Hypercube Sampling (LHS) to generate an initial population with minimum correlation between samples [44]. Thereafter, for each particle, the OF and the constraints are evaluated defining the level of violation of each constraint. Each particle is addressed to a specific aim according to their violation value. If none of the constraints are violated, this particle is labelled as feasible, and it will be addressed to minimize the objective function. Otherwise, if it violates at least one constraint, it is labelled as unfeasible, and it will try to find the right path to minimize the constraint violation. If more than one constraint is violated, only the maximum violation is considered at that point. Therefore, it is possible to assume that each particle is able to see only the envelope of the maximum violations for all points in the solution space. For this reason, the current approach has been named as a "multi-strategy" PSO. In this way, it is not necessary to define some

arbitrary violation penalty factor because the code directly relies on the envelope of the violation of the constraints in a particle position at a certain iteration number. After the first population is randomly sampled and evaluated, the role and the aim of each particle have been defined, and the swarm evolution cycle can start, as illustrated in Figure 3. The evolutionary phase of the PSO involves the Velocity update according to the before mentioned formulation (4) and the Position update according to Equation (5). After that, the cognitive memory (pbest) of each particle is updated if a better feasible position is reached with respect to the previous iterations, and the local best attractor (lbest) and the best position for the current generation (gbest) are also updated. The termination criterion is encountered when a predefined maximum number ($k_{\max}$) of iterations is reached.

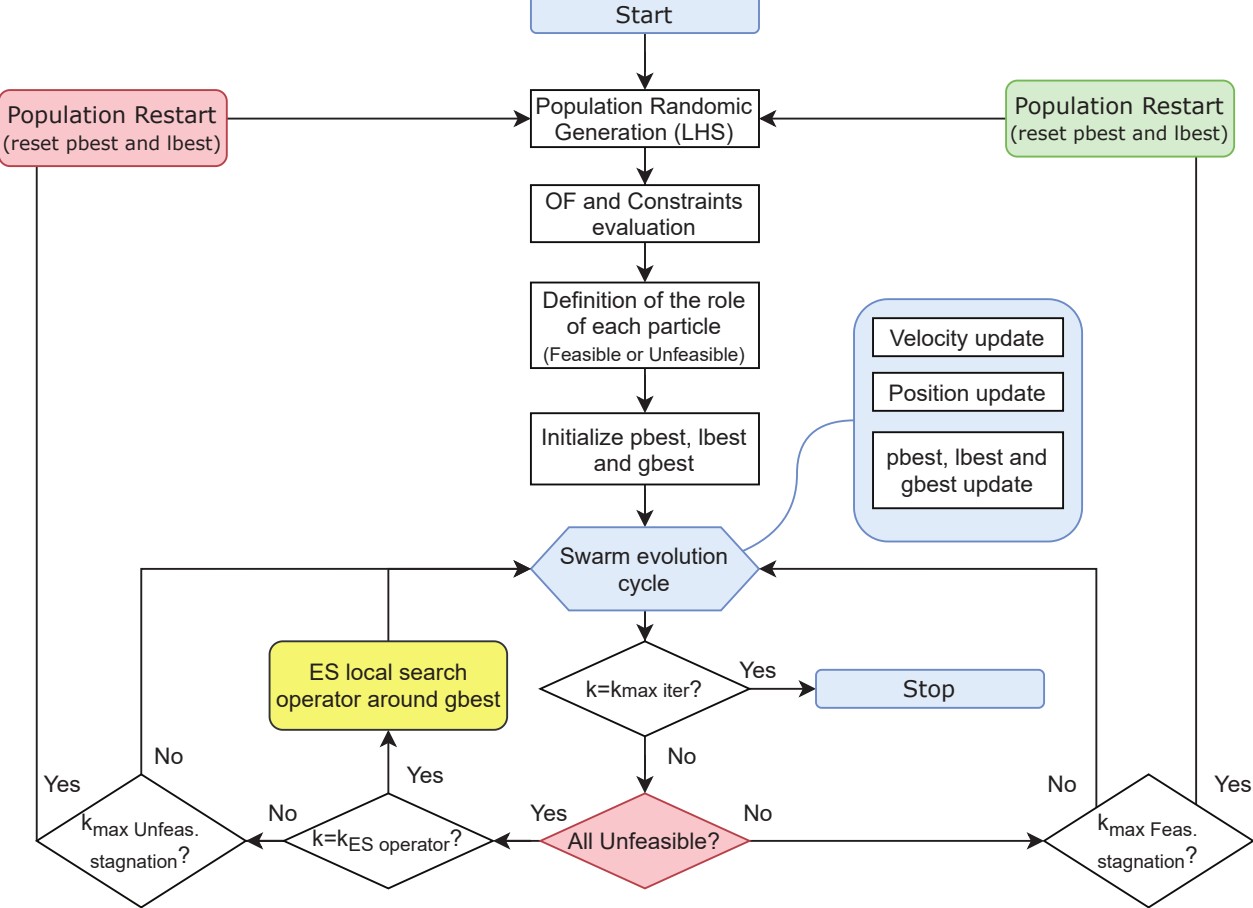

**Figure 3.** Enhanced PSO multi-strategy flowchart.

It may happen that the feasible region is quite little and narrow with respect to the entire search space; therefore, after some iterations, the swarm also may not have found the feasible region yet. Since the swarm has so far minimized the constraint violation, the swam has probably converged to an unfeasible point with the minimum value of constraint violation, and the feasible region may be located relatively close to that point. This fact suggests that by enhancing the local exploration around the so far unfeasible gbest founded point, the algorithm could be able to identify the feasible search space. Therefore, if the swarm has stalled to an unfeasible point for a number $k = k_{\text{ES operator}}$ of iterations, a local search operator based on the Evolutionary Strategy approach is thus performed. The Evolutionary Strategy (ES) algorithm is another famous paradigm of the classical EAs based on Darwinian Selection and it was developed by Ingo Rechenberg and Hans-Paul Schwefel at the Technical University of Berlin around the 1960s [1,45]. Without entering deeper into the details of this algorithm, it is necessary to recall that this is a population-based method which relies on the survival of the fittest members. Starting from a parent population,

the best individuals have a greater chance to be selected and evolve, forming a certain number of offspring which are generated throughout a slight mutation in the genome of the selected parents. The degree of mutation is governed by a mutation step, which is usually drawn by a Gaussian normal distribution $N(0, \sigma)$, in which $\sigma$ is also known as the *mutation step* size [45,46]. In formulae, it is possible to express that each gene of a selected parent $x_i$ undergoes a mutation procedure which produces a new offspring's gene equal to $x_i + N(0, \sigma)$. Then, the parents and the offspring will compete for survival, and only the best individuals will survive to the next generation. The main advantage of ES is that it is based on a single parameter to be tuned, the mutation step $\sigma$. Many variants of ES were developed in recent decades as mentioned in [45], but the self-adaptation strategy (also denoted as $\sigma$SA-ES or simply SA-ES [47–49]) is taken into account in the current study. To perform an SA-ES, it is necessary to consider a new representation for the individuals. From a practical point of view, when the parent genome is slightly mutated, if the generated offspring is better in terms of OF evaluation, this offspring will probably survive to the next generation, and it will probably spread its improved genome in the next iterations. Based on this observation, the mutation step can also be added to the original genome of the parent chromosome, giving a new individual representation such as $(x_1, \ldots, x_n, \sigma)$. In this way, not only the genes but also the mutation step undergoes the mutation operator. Thus, if a better offspring is obtained, it will survive and spread its chromosome information, which now implicitly takes into account a new adaptive mutation step. Therefore, in an indirect manner, good individuals will also generate good mutation steps which will be adaptively tuned during the next generations. The above-mentioned approach is known in the literature as SA-ES with uncorrelated mutation with one step size [46,48]. When a number of different mutation steps are considered, one for each gene in the chromosome, such as $(x_1, \ldots, x_n, \sigma_1, \ldots, \sigma_n)$, the adaptive ES strategy is called SA-ES with uncorrelated mutation with $n$ step size [46,48]. It is now clear that the main advantage to introduce the ES local search operator to the current enhanced PSO implementation is due to the fact that it can be implemented without manually tuning other parameters because they are self-tuned by the algorithm itself. For example, in [50], a hybridization of the PSO with ES was performed to enhance the classical velocity update with an adaptive update of the inertia weight and the acceleration factors. For the sake of completeness, there are more sophisticated self-adaptive approaches which take into account also the correlations among the various step sizes associated with the various genes, which are named as SA-ES with correlated mutation [46,48] or covariance matrix adaptation CMA-ES [47,48,51]. In the current study, the SA-ES with uncorrelated mutation with $n$ step size operator is integrated with the PSO inside a local search operator in order to try to locate the feasible region if the swarm stalls to an unfeasible point for $k_{\text{ES operator}} = 10$ iterations. From the unfeasible gbest starting point $x^{Gb,\text{unfea}}$, a population of $N_p = 50$ parent points is sampled from a multivariate Gaussian mixture model in which each component has mean equal to the gbest's $i$-th component, $x_i^{Gb,\text{unfea}}$, and covariance equal to a first attempt mutation step $\sigma_i$. Each $i$-th mutation step is defined by:

$$\sigma_i = |\tau \cdot N(0,1)| \tag{15}$$

i.e., the absolute value of the product of a random number sampled from a normal standard distribution $N(0,1)$ multiplied to a learning rate parameter $\tau$, which is suggested in [47] to be assumed as $1/\sqrt{N_p}$. Then, a first population of $N_o = 100$ mutated offspring is generated by randomly selected parents adopting a mutation scheme in which the $i$-th new mutation step size component is updated as:

$$\sigma_{i,\text{off}} = \max(0, |\sigma_i + N(0,1)|). \tag{16}$$

Thereafter, a new offspring point is obtained by adding to the parent position the mutated vector sampled by the multivariate Gaussian mixture model with a mean equal to a zero array and covariance equal to the mutation step size vector updated as above. Subsequently,

the mutated offspring are added to the parent population, and the best $N_p$ individuals are selected to survive to the next iteration in terms of constraints violations (or in case of feasible points in term of OF). In the ES jargon, this approach is called the $\mu + \lambda - ES$ strategy because the $\mu$ ($N_p$) parents will compete with both each other and also new $\lambda$ ($N_o$) offspring, but finally, only $\mu$ individuals will survive, whereas the others will be discarded [47]. This mechanism resembles the steady-state approach of other EAs likewise in the genetic algorithm GA [1]. The ES operator could theoretically perform a maximum number of local iterations equal to $k_{max,Local} = 50$, but in the case that a feasible point is found, the ES evolutionary cycle is interrupted. This new feasible point is thus set up as the gbest of the previous PSO swarm, which remained in a sort of standby state while the local ES operator was in action. In summary, the PSO cycle, which has entered in the ES operator due to the fact that it stalled for $k_{ES\ operator} = 10$ iterations on an unfeasible gbest point, can now restart again as usual with an improved knowledge provided by a new feasible posed gbest point found by the local search ES operator. The numerical example Problem g06, whose statement is in the Appendix A (Sickle Problem [52]), has been depicted in Figures 4 and 5 to graphically show the enhanced multi-strategy PSO procedure. Each swarm particle is able to see only the sub-figures (a), (c), and (e) of Figure 4 when its position is inside the feasible region (with the role to minimize the OF); otherwise, it is able to see only the landscape produced by the constraint envelope, subfigures (b), (d), and (f) of Figure 4. After 10 stagnations on the unfeasible gbest point (black cross in sub-figures (a), (b), (c), and (d) of Figure 4), the ES operator was performed. It generated a local population of points near the unfeasible gbest point, which are colored as purple if they are unfeasible or green if they are feasible. Then, this population evolves with the before explained SA-ES approach until at least one point falls inside the feasible region (which is the space between the two blue parabolas) or the maximum number of local iterations is reached. In that specific case, at the first local iteration, some feasible points were already found. Therefore, the best individuals in term of OFs was selected among the green points of Figure 4c,d, and then the PSO could continue its evolutionary cycles until the maximum number of iterations were reached ($k_{max} = 500$). The history of the optimal solution found during the PSO iterations is depicted in Figure 5.

For some very hard problems, it may also happen that after the action of the ES local search operator, the feasible region is not found. In that case, the PSO starts the evolution cycle again with the same unfeasible gbest point for some other iterations until the feasible region is found. Otherwise, when the iterations reach a total number of unfeasible stagnations $k_{max\ Unfeas\ Stagn} = 15$, the complete reset of the population is performed. In practise, the algorithm completely restarts again from the first point of the flowchart, as shown in Figure 3. Therefore, the hope is that a completely new random sampling of the initial swarm will generate a new initial configuration which may find this time the right path to the optimal solution of the optimization problem.

On the contrary, when the PSO normally finds the feasible region and it optimizes the solution until it reaches a gbest which stagnates for a certain number of iterations $k_{max\ Feas\ Stagn} = 50$, the population is restarted as well. This is due to the fact that the so far found optimal solution could be a local minimum. If there is a certain number of iterations left before reaching the maximum PSO available iterations, $k < k_{max}$, the swarm is thus restarted again from the first step of the PSO flowchart. In that case, all the memories of the population are reset (pbests and lbests), but the so far found optimal solution (gbest) remains unchanged, unless a better solution in terms of OF is found from the new restarted-swarm exploration phase.

In the following section, the enhanced multi-strategy PSO has been tested on some constrained numerical benchmark literature problems, and the results are compared with two PSO implementations, which adopt a typical penalty approach.

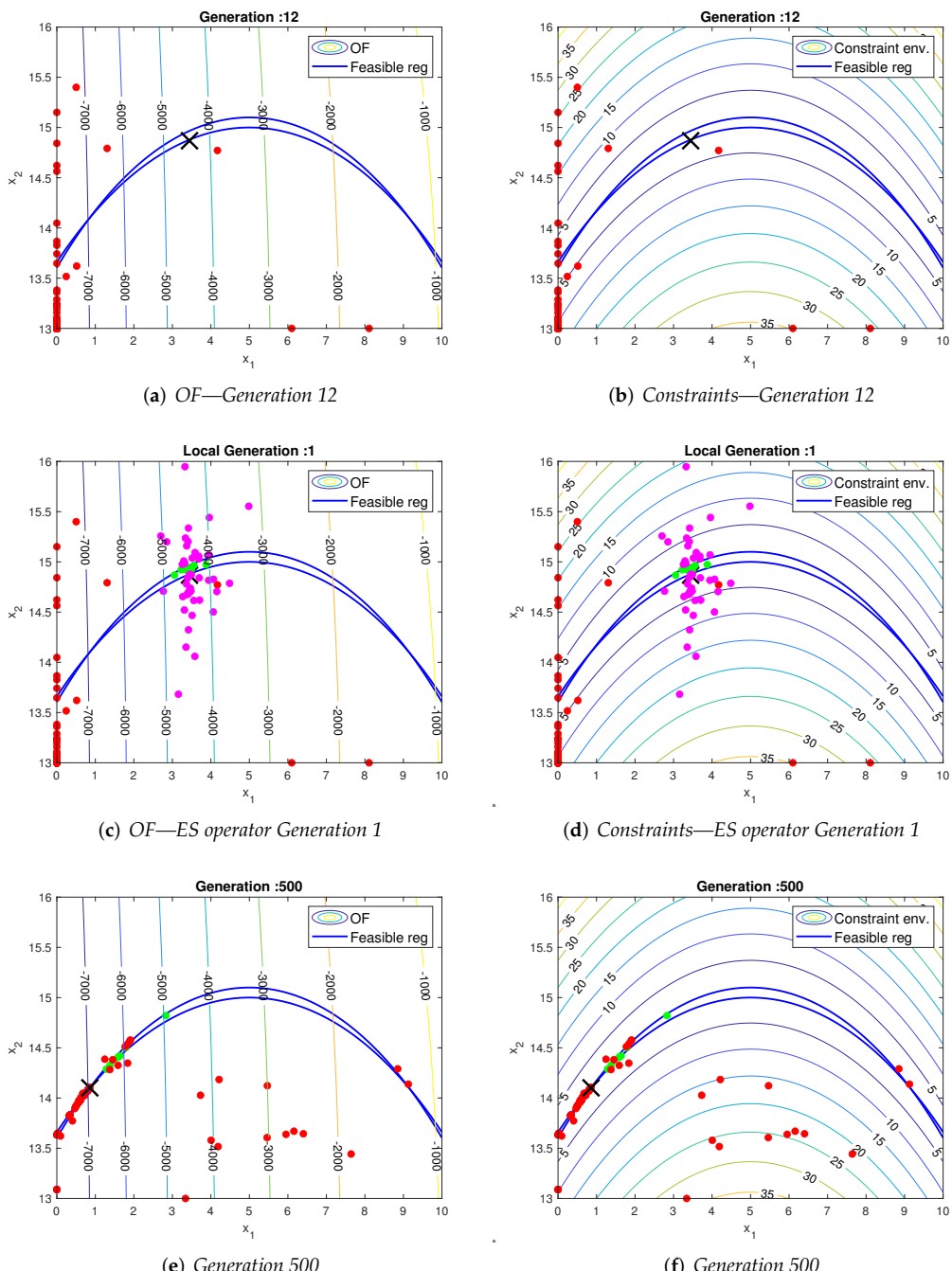

(**a**) *OF—Generation 12*

(**b**) *Constraints—Generation 12*

(**c**) *OF—ES operator Generation 1*

(**d**) *Constraints—ES operator Generation 1*

(**e**) *Generation 500*

(**f**) *Generation 500*

**Figure 4.** Example Problem g06, see the Appendix A (Sickle Problem [52]); (**a**,**b**) the OF and constraints envelope contour representations, respectively at generation 12. The black cross marker is the unfeasible gbest, the red dots are the swarm points. (**c**,**d**) After 10 unfeasible stagnations, the ES local search operator generate a local search population (purple dots) to find the feasible region (green dots). (**e**,**f**) the OF and constraints envelope contour representations, respectively, at the final generation 500. The black cross marker is the feasible gbest point, the red ones are the particles in a unfeasible region, and the green ones are the particle inside the feasible region.

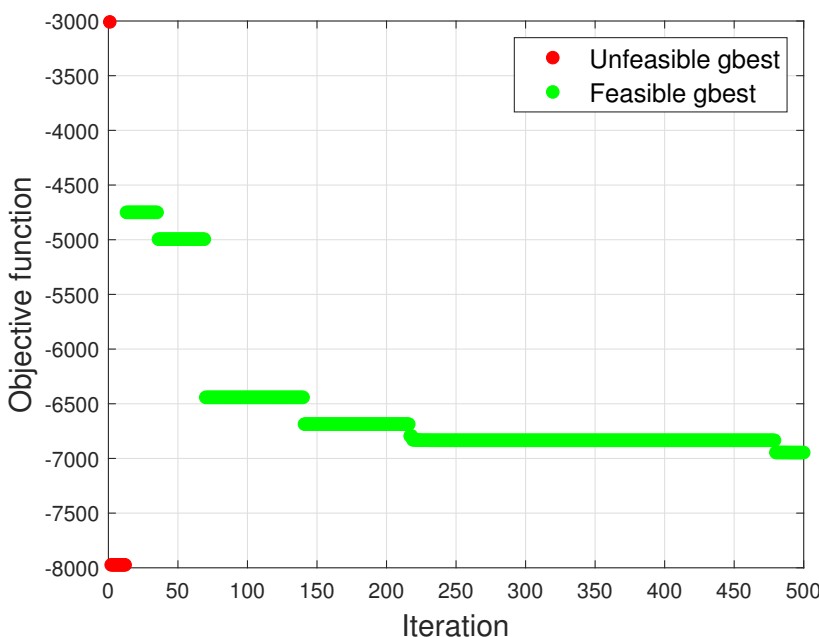

**Figure 5.** Example Problem g06, see Appendix A (Sickle Problem [52]); Objective function history of the gbest (optimal solution).

## 4. Numerical Test and Comparisons

The new enhanced multi-strategy PSO illustrated in the previous section was implemented in a Matlab environment and some numerical constrained benchmark tests from the literature were analysed. In particular, the statements of the mathematical constrained problems were taken from [53], in which a total of 13 constrained problems are illustrated. In the current study, only some problems were considered, in particular, the problems with inequalities constraints only were analysed. As stated before, the PSO does not perform very well with equality constraints despite some strategies being proposed in literature to convert each equality constraint into a couple of equivalent inequality constraints. For the sake of completeness, the selected problem statements are also reported in the Appendix A of the present paper. In order to make some comparisons with the other more classical constraint handling approaches, the current enhanced multi-strategy PSO is compared with a more classic penalty approach. For this purpose, the PSO code proposed by [54] was adopted and modified in order to take into consideration both a static penalty approach as previously mentioned in (6) and also with a dynamic penalty as in (9). The penalty factors were properly tuned problem by problem in order to obtain the optimal results. The swarm size was set to $N = 100$, and the maximum allowable iterations were fixed to $k_{\max} = 500$ for all the PSOs considered. The comparisons shown in Table 1 are developed from the results obtained by 50 independent runs and making comparisons among best and worst results and the mean and standard deviation of the OF from the dataset of the 50 final results for the 3 different PSOs. The results in Table 1 produced by the enhanced multi-strategy PSO are satisfactory for the selected numerical problems, and they are generally consistent if compared with the theoretical results and with the other penalty-based PSO implementations. This proves the effectiveness of the current enhanced PSO implementation to deal with constrained optimization problems without the tedious calibration of too many arbitrary parameters. Because of these initial promising results, future works should therefore include some other numerical applications and some engineering practical optimization problems.

**Table 1.** Selected numerical benchmark examples taken from [53] and comparisons of the final results for 50 runs among the enhanced multi-strategy PSO (*PSO_MS*), the PSO with static penalty (*PSO_ST*), and the PSO with dynamic penalty (*PSO_DYN*).

| **Problem g01** | *PSO_MS* | *PSO_ST* | *PSO_DYN* |
|---|---|---|---|
| *optimum* | | −15.000 | |
| *best OF* | −15.000 | −15.000 | −15.0 |
| *worst OF* | −12.002 | −12.000 | −12.000 |
| *mean* | −14.443 | −13.938 | −13.920 |
| *std* | 0.89478 | 1.4333 | 1.4546 |
| **Problem g02** | *PSO_MS* | *PSO_ST* | *PSO_DYN* |
| *optimum* | | 0.803619 | |
| *best OF* | 0.80357 | 0.80146 | 0.79358 |
| *worst OF* | 0.60963 | 0.52013 | 0.38285 |
| *mean* | 0.75896 | 0.70105 | 0.66597 |
| *std* | 0.063604 | 0.07356 | 0.087006 |
| **Problem g04** | *PSO_MS* | *PSO_ST* | *PSO_DYN* |
| *optimum* | | −30,665.539 | |
| *best OF* | −30,666.0 | −30,666.0 | −31,207.0 |
| *worst OF* | −30,666.0 | −30,665.0 | −30,137.0 |
| *mean* | −30,666.0 | −30,665.0 | −31,138.2 |
| *std* | 2.20e-05 | 0.86587 | 252.2036 |
| **Problem g06** | *PSO_MS* | *PSO_ST* | *PSO_DYN* |
| *optimum* | | −6961.81388 | |
| *best OF* | −6961.8 | −6973.0 | −6963.0 |
| *worst OF* | −6958.4 | −6973.0 | −6963.0 |
| *mean* | −6960.7 | −6973.0 | −6963.0 |
| *std* | 0.97521 | 0.0000 | 0.0000 |
| **Problem g07** | *PSO_MS* | *PSO_ST* | *PSO_DYN* |
| *optimum* | | 24.3062091 | |
| *best OF* | 24.426 | 25.034 | 24.477 |
| *worst OF* | 27.636 | 30.203 | 30.112 |
| *mean* | 25.4129 | 28.508 | 27.043 |
| *std* | 1.1209 | 1.4351 | 1.8821 |
| **Problem g08** | *PSO_MS* | *PSO_ST* | *PSO_DYN* |
| *optimum* | | 0.095825 | |
| *best OF* | 0.095825 | 0.095825 | 0.095825 |
| *worst OF* | 0.095825 | 0.095825 | 0.095825 |
| *mean* | 0.095825 | 0.095825 | 0.095825 |
| *std* | 6.96e-17 | 6.77e-17 | 7.10e-17 |
| **Problem g09** | *PSO_MS* | *PSO_ST* | *PSO_DYN* |
| *optimum* | | 680.6300573 | |
| *best OF* | 680.64 | 680.63 | 680.63 |
| *worst OF* | 680.98 | 680.72 | 680.73 |
| *mean* | 680.73 | 680.66 | 680.66 |
| *std* | 0.079365 | 0.017526 | 0.018915 |

**Table 1.** *Cont.*

| Problem g12 | PSO_MS | PSO_ST | PSO_DYN |
|---|---|---|---|
| *optimum* | | 1.0 | |
| *best OF* | 1.0 | 1.0 | 1.0 |
| *worst OF* | 1.0 | 1.0 | 1.0 |
| *mean* | 1.0 | 1.0 | 1.0 |
| *std* | 0.0000 | 2.12e-15 | 0.0000 |

## 5. Structural Optimization on Literature Benchmarks

In this final part, some well-acknowledged structural engineering optimization problems from the literature have been adopted for evaluating the performances of the proposed multi-strategy PSO algorithm with the unfeasible local search operator. In the analysed benchmarks, the multi-strategy PSO has been compared with other optimization strategies, i.e., the PSO with static and dynamic penalty inspired by the code of [54] and with the GA from Matlab's built-in code functions. Structural optimization problems can be mainly grouped into three main categories [55]: the *size optimization*, where the aim is to find the optimal size of the structural elements; the *shape optimization*, in which the design variables govern the structural shape; the *topology optimization*, which is the more complex because it involves the modification of the structural typology and morphology. These problems could be tackled separately or even combined. Mainly focusing on the contribution of [56], in the current study, three different truss design constrained size optimization problems have been analysed. The main goal of truss design problems is to minimize the total weight $w$ of the structure, which is indirectly connected to the material consumption volume amount and thus to the cost of the structure [55]. Indeed, adopting a certain material with unit weight $\rho_i$, the main goal results in seeking for the optimal cross-sectional areas $A_i$ to be devoted to every structural element in the design domain. A first constraint is represented by the box-constraint related to the admissible range of cross section area values to be adopted $A_i \in [A_i^{\mathrm{LB}}, A_i^{\mathrm{UB}}]$. Thereafter, at least two other inequality constraints have to be considered. The first one is related to the respectfulness of the maximum allowable stress $\sigma_{\mathrm{adm}}$ in each truss member (resistance-side constraint) and the second one is referred to the respectfulness of a maximum displacement threshold $\delta_{\mathrm{adm}}$ (deformation-side constraint). The general formulation of the truss design problem can be stated as follows:

$$
\begin{aligned}
\min_{\boldsymbol{x} \in \Omega} \quad & f(\boldsymbol{x}) = \sum_{i=1}^{N_e l} \rho_i L_i A_i \\
\text{s.t.} \quad & A_i^{\mathrm{LB}} \leq A_i \leq A_i^{\mathrm{UB}} \\
& \sigma_i \leq \sigma_{\mathrm{adm}} \\
& \delta \leq \delta_{\mathrm{adm}}
\end{aligned}
\tag{17}
$$

where $N_e l$ is the total number of elements in the truss design domain and $L_i$ is the actual length of each member. The material adopted in the current study is structural steel with unit weight of $\rho_i = \rho = 0.1 \ \mathrm{lb/in}^3$ (1 lb/in$^3$ is equal to 0.0276799 kg/cm$^3$) and Young's modulus of $10^7$ psi (1 psi is equal to 0.00689476 MPa).

### 5.1. Ten-Bar Truss Design Optimization

The first problem analysed is referred to as a 10 bar truss cantilever structure, as depicted in Figure 6. In the cantilever structure, each member has been labelled with a number from 1 to 10. The cantilever span is in total 720 inches (1 inch is equal to 25.4 mm), and the depth is 360 in. The truss structure is loaded by 2 downward forces of 100 kips each (1 kips is equal to 4.4482 kN). The design vector considers cross-section areas as continuous variables belonging to the a close interval $[0.1, 35]$ in$^2$. The maximum allowable deflection both in horizontal and vertical direction for every node has been set to

$\delta_{\mathrm{adm}} = \pm 2$ in, whereas the maximum allowable stress is equal to $\sigma_{\mathrm{adm}} = \pm 25$ ksi. In total, 100 independent executions have been performed, and the mean and standard deviation of the OFs have been calculated. A population size of 50 particles and a maximum iterations number of 500 have been set both for the multi-strategy PSO and the GA. For the PSO with penalty approaches, 500 particles have been set as the swarm size because of their very poor results when only 50 particles have been considered. The optimization results obtained are reported in Table 2, which compares the multi-strategy PSO with the PSO with static penalty (PSO-Static), with dynamic penalty (PSO-Dynamic), and with GA. It is worth noting that the penalty approaches fail dreadfully, in this case, to deal with real-life structural design problems, whereas the proposed multi-strategy PSO algorithm produces good results which are comparable with the GA and quite close to the actual unknown optimum solution.

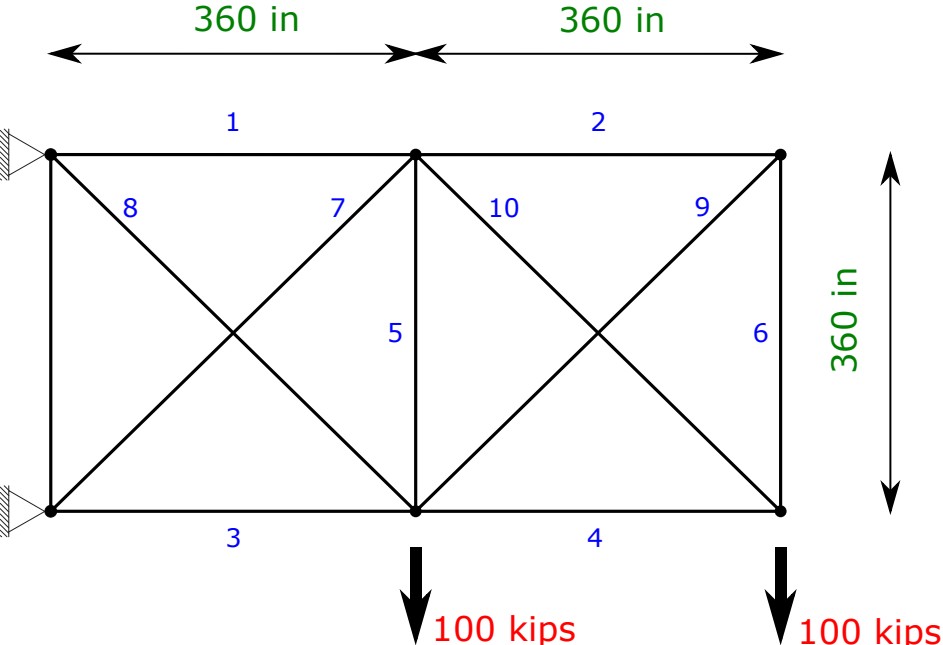

**Figure 6.** Graphical representation of the 10 bar truss design optimization problem.

*5.2. Twenty-Five-Bar Truss Design Optimization with Multi-Load Cases Conditions*

The second structural optimization problem analysed is referred to as the 25 bar three-dimensional truss tower structure, as depicted in Figure 7. In plan view, the tower footprint is a square of side 200 in, which tapers to 75 in at an elevation of 100 in, and finally reaches the maximum elevation at 200 in from the ground. The structural nodes have been labelled with a number from 1 to 10. The design vector considers the cross section areas of each member as continuous variables belonging to the close interval $[0.01, 3.40]$ in$^2$. The cross-sectional areas have been gathered into eight groups, as depicted in Figure 8, in order to reduce the dimensionality of the design vector. The maximum allowable displacement has been set to $\delta_{\mathrm{adm}} = \pm 0.35$ in in every direction, whereas the maximum allowable stress of each member has been to $\sigma_{\mathrm{adm}} = \pm 40$ ksi. Furthermore, the current structural problem takes into account two different load cases during the optimization process, as shown in Figure 7. In total, 100 independent executions have been performed, and the mean and standard deviation of the OFs have been calculated. A population size of 50 particles and a maximum iterations number of 500 have been set both for the multi-strategy PSO and the GA. For the PSO with penalty approaches, 500 particles have been set as the swarm size because of their very poor results when only 50 particles have been considered. The optimization results obtained are reported in Table 3, which compares the multi-strategy PSO with the PSO with the static penalty (PSO-Static), with the dynamic penalty (PSO-Dynamic), and with the GA. It is worth noting that, even in this case, the penalty

approaches dreadfully fail to deal with real-life structural design problems, whereas the proposed multi-strategy PSO algorithm produces good results which are comparable with the GA and quite close to the actual optimum solution.

**Table 2.** Ten-bar truss design example: results comparisons for 100 runs among the enhanced multi-strategy PSO (*PSO-MS*), the PSO with static penalty (*PSO-Static*), and the PSO with dynamic penalty (*PSO-Dynamic*) and GA.

| | Cross-Section [in²] | | | | |
|---|---|---|---|---|---|
| **Element** | **Ref. Sol. from [56]** | **PSO-Static** | **PSO-Dynamic** | **GA** | **PSO-MS** |
| **1** | 28.920 | 29.6888 | 30.3092 | 30.145 | 30.372 |
| **2** | 0.100 | 18.3211 | 14.7464 | 0.100 | 0.110 |
| **3** | 24.070 | 19.9891 | 16.5717 | 22.466 | 23.644 |
| **4** | 13.960 | 18.2381 | 25.1945 | 15.112 | 15.391 |
| **5** | 0.100 | 2.3404 | 4.5489 | 0.101 | 0.101 |
| **6** | 0.560 | 20.8674 | 26.1207 | 0.543 | 0.496 |
| **7** | 21.950 | 21.1805 | 32.2698 | 21.667 | 20.984 |
| **8** | 7.690 | 16.0851 | 0.2168 | 7.577 | 7.410 |
| **9** | 0.100 | 6.0845 | 7.5871 | 0.100 | 0.103 |
| **10** | 22.090 | 25.5632 | 23.524 | 21.695 | 21.378 |
| **best OF** [lb] | 5076.310 | 6141.986 | 6333.035068 | 5063.250 | 5063.328 |
| **worse OF** [lb] | - | 8415.134 | 8675.749551 | 5144.148 | 5229.108 |
| **mean** [lb] | - | 7294.455 | 7501.394582 | 5079.744 | 5076.473 |
| **std. dev.** [lb] | - | 516.7823 | 475.3885728 | 14.1194 | 24.8666 |

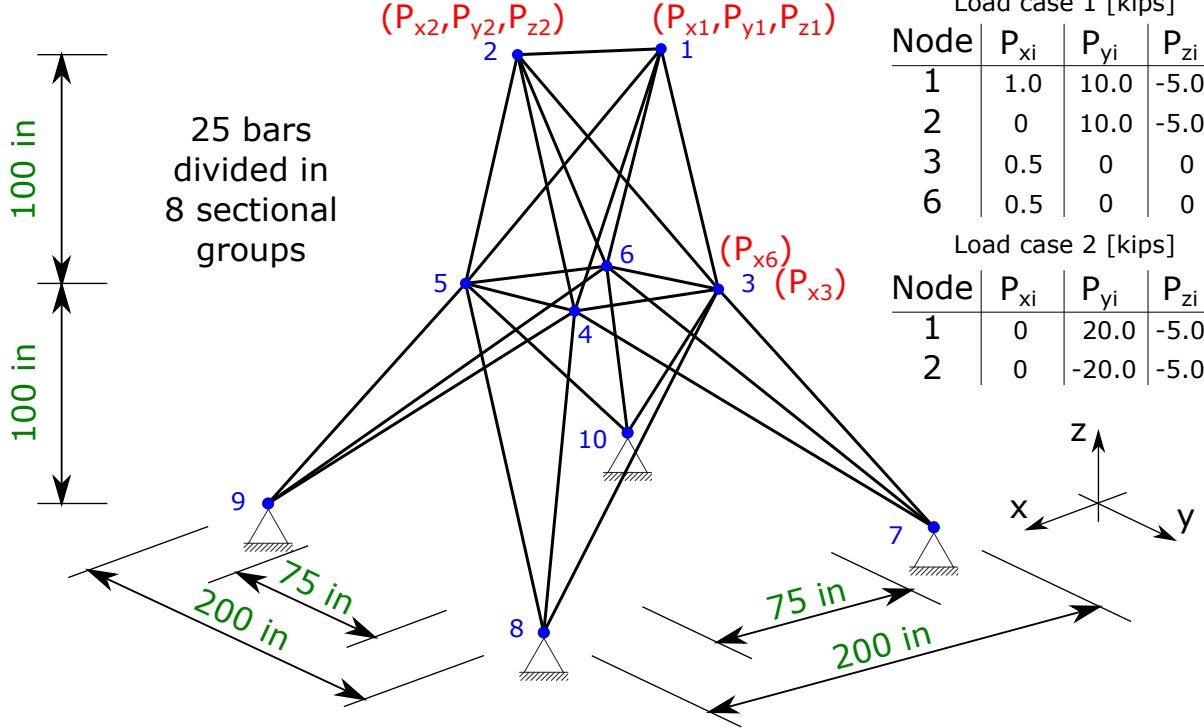

**Figure 7.** Graphical representation of the 25 bar truss design optimization problem.

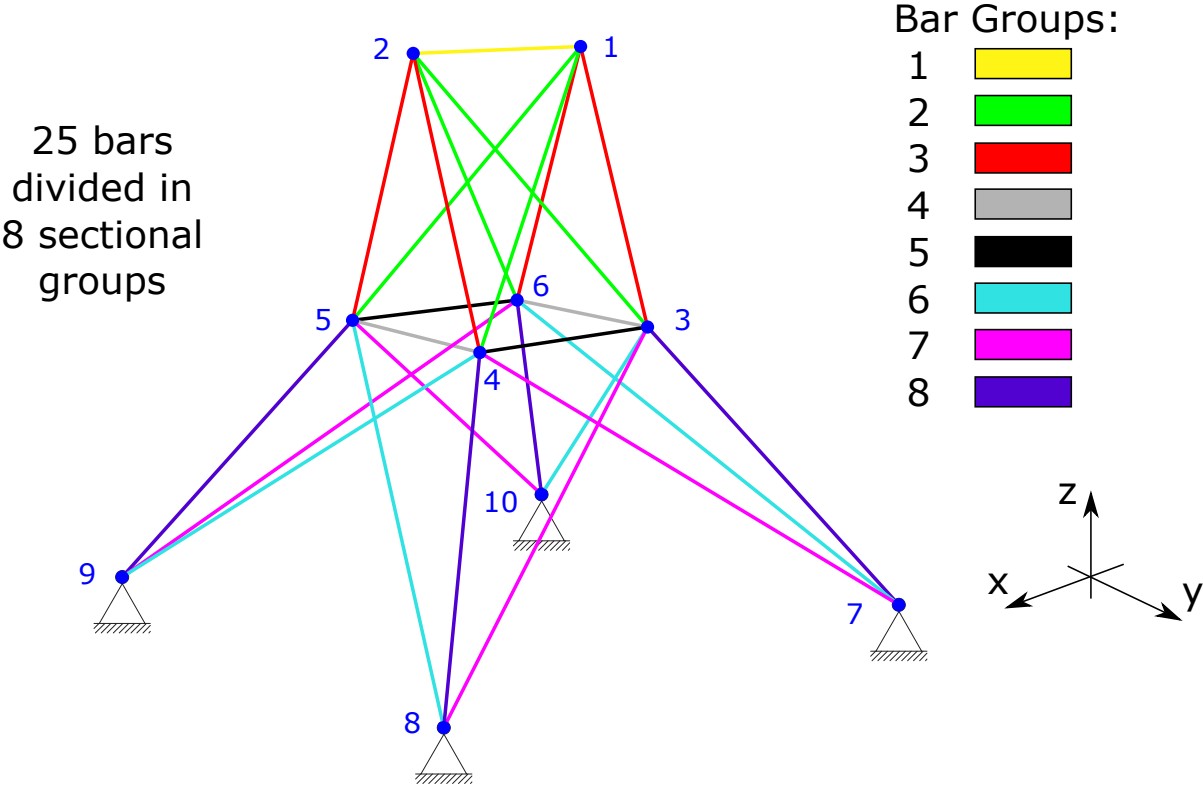

**Figure 8.** Graphical representation of the 8 bar groups in which are collected all the members of the 25 bar truss design optimization problem.

**Table 3.** Twenty-five bar truss design example: results comparisons for 100 runs among the enhanced multi-strategy PSO (*PSO-MS*), the PSO with static penalty (*PSO-Static*), and the PSO with dynamic penalty (*PSO-Dynamic*) and GA.

| Cross-Section [in$^2$] | | | | | |
|---|---|---|---|---|---|
| **Bar Group** | **Ref. Sol. from [56]** | **PSO-Static** | **PSO-Dynamic** | **GA** | **PSO-MS** |
| **1** | 0.100 | 2.054 | 1.116 | 0.010 | 0.011 |
| **2** | 1.800 | 2.675 | 2.670 | 2.023 | 1.976 |
| **3** | 2.300 | 1.402 | 1.942 | 2.941 | 2.989 |
| **4** | 0.200 | 3.388 | 0.166 | 0.010 | 0.010 |
| **5** | 0.100 | 0.204 | 0.342 | 0.010 | 0.011 |
| **6** | 0.800 | 0.453 | 1.985 | 0.671 | 0.690 |
| **7** | 1.800 | 1.274 | 1.976 | 1.673 | 1.689 |
| **8** | 3.000 | 0.048 | 2.345 | 2.694 | 2.654 |
| **best OF** [lb] | 546.010 | 568.186 | 596.058 | 545.236 | 545.249 |
| **worse OF** [lb] | - | 100,583.118 | 22,954.297 | 557.755 | 552.378 |
| **mean** [lb] | - | 1673.393 | 1122.518 | 547.828 | 546.003 |
| **std. dev.** [lb] | - | 9991.0201 | 3129.3192 | 2.0743 | 0.7879 |

### 5.3. Seventy-Two-Bar Truss Design Optimization with Multi-Load Cases Conditions

The last structural optimization problem analysed in the current study is referred to as a 72 bar three-dimensional truss tower structure, as depicted in Figure 9. In plan view, the tower footprint is a square of side 120 in, with 4 modular floors, each of them with a height of 60 in. The structural nodes have been labelled with a number from 1 to 20. The design vector considers the cross-sectional areas of each member as continuous variables belonging to the close interval $[0.1, 3.0]$ in$^2$. There are 18 bars inside each modular floor which can

be grouped in 4 groups, as depicted in Figure 10. Therefore, since there are 4 floors, the cross-sectional areas have been parametrized into 16 groups in total in order to reduce the dimensionality of the design vector. The maximum allowable displacement has been set to $\delta_{adm} = \pm 0.25$ in in every direction, whereas the maximum allowable stress of each member has been set to $\sigma_{adm} = \pm 25$ ksi. Furthermore, the current structural problem takes into account two different load cases during the optimization process, as shown in Figure 9. In total, 100 independent executions have been performed, and the mean and standard deviation of the OFs have been calculated. A population size of 50 particles and a maximum iterations number of 500 have been set both for the multi-strategy PSO and the GA. For the PSO with penalty approaches, 500 particles have been set as the swarm size because of their very poor results when only 50 particles have been considered. The optimization results obtained are reported in Table 4, which compares the multi-strategy PSO with the PSO with the static penalty (PSO-Static), with the dynamic penalty (PSO-Dynamic), and with the GA. Similarly to the previous cases, it is worth noting that the penalty approaches dreadfully fail to deal with real-life truss design structural optimization problems, whereas the proposed multi-strategy PSO algorithm produces good results which are comparable with the GA and quite close to the actual optimum solution. It is worth noting that the mean value and the best one are very close to the reference optimal solution from [56]. The final solution is even characterized by a low standard deviation among the 100 algorithm runs, demonstrating that the multi-strategy PSO is able to reach the optimal results in a more reliable way, reducing the uncertainties and scattering of the final results.

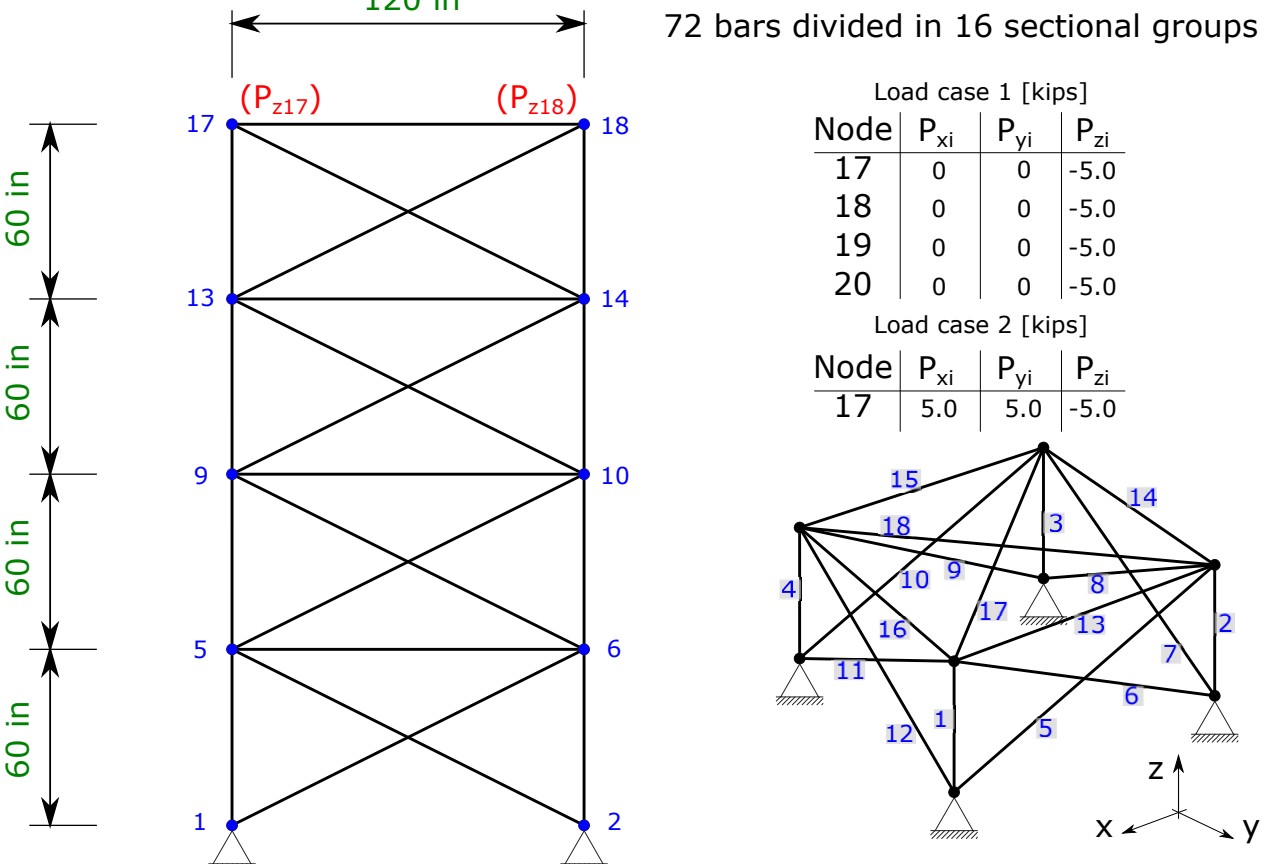

**Figure 9.** Graphical representation of the seventy-two bars truss design optimization problem.

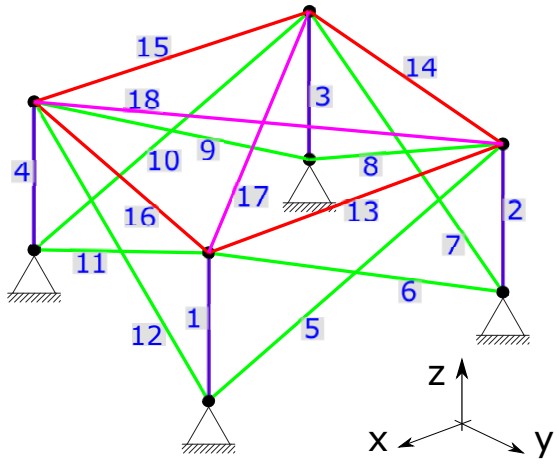

**Figure 10.** Graphical representation of the four bar groups in which are collected the members inside one module of the seventy-two bars truss design optimization problem.

**Table 4.** Seventy-two bars truss design example: results comparisons for 100 runs among the enhanced multi-strategy PSO (*PSO-MS*), the PSO with static penalty (*PSO-Static*) and the PSO with dynamic penalty (*PSO-Dynamic*) and GA.

| | Cross-Section $[\text{in}^2]$ | | | | |
|---|---|---|---|---|---|
| **Bar Group** | **Ref. Sol. from [56]** | **PSO-Static** | **PSO-Dynamic** | **GA** | **PSO-MS** |
| **1** | 2.026 | 2.176 | 0.746 | 1.801 | 1.856 |
| **2** | 0.533 | 0.661 | 0.539 | 0.545 | 0.523 |
| **3** | 0.100 | 2.686 | 0.523 | 0.100 | 0.100 |
| **4** | 0.100 | 1.771 | 2.660 | 0.100 | 0.100 |
| **5** | 1.157 | 1.662 | 2.316 | 1.311 | 1.301 |
| **6** | 0.569 | 0.276 | 1.051 | 0.511 | 0.519 |
| **7** | 0.100 | 0.158 | 0.642 | 0.100 | 0.100 |
| **8** | 0.100 | 0.986 | 2.370 | 0.100 | 0.100 |
| **9** | 0.514 | 0.271 | 0.757 | 0.531 | 0.539 |
| **10** | 0.479 | 1.240 | 0.793 | 0.520 | 0.507 |
| **11** | 0.100 | 0.517 | 0.453 | 0.100 | 0.100 |
| **12** | 0.100 | 0.378 | 1.754 | 0.107 | 0.101 |
| **13** | 0.158 | 0.119 | 2.236 | 0.157 | 0.157 |
| **14** | 0.550 | 0.794 | 1.677 | 0.534 | 0.540 |
| **15** | 0.345 | 1.363 | 0.824 | 0.386 | 0.403 |
| **16** | 0.498 | 1.190 | 0.830 | 0.561 | 0.564 |
| **best OF** [lb] | 379.310 | 629.108 | 662.148 | 380.150 | 379.753 |
| **worse OF** [lb] | - | 1054.764 | 1110.795 | 400.147 | 381.541 |
| **mean** [lb] | - | 874.024 | 854.233 | 383.377 | 380.150 |
| **std. dev.** [lb] | - | 88.8254 | 82.1187 | 3.7299 | 0.2766 |

## 6. Discussion

In the previous sections, it has been demonstrated that the proposed multi-strategy PSO algorithm provided quite interesting results. Foremost, focusing on numerical benchmark problems, the multi-strategy PSO technique has been compared with two other traditional PSO implementations which adopt the penalty function approaches to deal with constraints. The three algorithms have been executed 50 independent times for each numerical problem stated in the Appendix A, and the final results have been collected in Table 1. The optimization results have been presented in terms of the best solution, the worst solution, the mean of the OF values, and the standard deviation of the final results.

These parameters evidence the scattering in the found solutions by the various algorithms. Specifically, the standard deviation parameter gives a direct insight into the degree of failure of the meta-heuristic algorithm to find the known benchmark solutions among the independent executions. In particular, the multi-strategy PSO presents in general lower values of a standard deviation compared with PSO-penalty methods, or at least the same order of magnitude. Furthermore, the multi-strategy PSO appears to be a more reliable algorithm because, focusing, e.g., on the problem g06, despite the standard deviation of the PSO-penalty being zero, they fail to reach the optimum solution. This fact highlights that, notwithstanding that the penalty functions method is very simple and easy to implemented, in general, it does not always represent the best approach to successfully deal with every kind of problem. Indeed, e.g., in problem g06, the nil value of the standard deviation actually points out how the penalty method provides a quite deterministic PSO algorithm which is trivially entrapped in the same local optimum among the independent runs, jeopardizing the potentialities of the stochastic search.

On the other hand, focusing on real-world engineering structural optimization problems, the multi-strategy PSO algorithm has revealed its powerful capabilities to deal with complex, combinatorially demanding, and highly non-linear optimization problems. For the sake of completeness, in these problems, a further comparison has been provided by the GA algorithm from the Matlab environment. This latter comparison is extremely relevant because it allows for performing a more objective evaluation which relies on a completely different implementation with respect to the PSO framework only. The optimization results of the 10 bar truss, 25 bar truss, and 72 bar truss problems have been reported in Tables 2–4, respectively. In all the analysed cases, the multi-strategy PSO provided very interesting results, which are really close or even better to the reference solution obtained from [56]. The penalty method revealed their weakness when dealing with these kinds of highly non-linear problems because they provided mean solutions quite far from the reference one and even more scattered when considering the standard deviation values. In conclusion, the proposed multi-strategy PSO algorithm provides an enhanced and more reliable implementation because it results in lower standard deviation values than the GA ones, at least in the last two problems hereby analysed, which are the most complex and computationally demanding.

## 7. Conclusions

The research and developments in the EAs field to solve optimization problems are continuously increasing because of their lack of mathematical proofs and also because the perfect algorithm to solve any kind of problem does not exist. Therefore, in the present study, a new variant of the PSO has been implemented for the purpose of studying a different way to deal with constrained optimization problems. In fact, the standard version of the PSO [17] lacked a proper mechanism to deal with constrained problems, and in literature [30,33,34], there are at least five main kinds of constraint-handling approaches. The so far most extensively used method in many different practical applications is the penalty function method. The main disadvantage of this technique is that it requires the user to tediously tune some arbitrary penalty factors, which is not always an easy task. In the current study, for the purpose of enhancing the performance of the standard version of the algorithm, the most important state-of-the-art improvements are also implemented, such as the inertia weight [23] and the neighbourhood topology [25]. Furthermore, in order to avoid a penalty-based approach, the violation degree of the constraints is directly exploited to define the aim of a particle which has to minimize this violation if it lies in the unfeasible region. Otherwise, if a particle lies in the feasible region, this particle is dedicated to minimize the OF. Another improvement is given by a local search self-adaptive ES operator, which takes action if the feasible region is not found by the PSO for a certain number of iterations. This allows the algorithm to spread the exploration around the so far unfeasible best solution found, which may be very close to the feasible region, if it is located in near this point. If the ES operator successfully finds the feasible region, this allows it

to boost the PSO, giving it an important hint on where the feasible region is located, as demonstrated in Figure 4. If the local operator fails to identify the feasible region, the swarm has probably been entrapped in a local unfeasible minimum quite far from the feasible region. Consequently, only a new randomly resampled swarm may probably find the right path to the feasible region and thus to the real optimum. This new enhanced PSO appears to be noticeably effective compared to other PSO algorithms which adopt a more traditional penalty-function-based method, as shown in Table 1. Outstanding results have been pointed out in the structural optimization benchmark analysed in the current study, which involves three truss design problems from the literature. The proposed PSO effectively dealt with real-life optimization problems, much better than traditional penalty approaches, and reached results comparable and competitive with other state-of-the-art implementations such as the GA.

Although the PSO algorithm already possesses two kinds of memories (cognitive and social), most of the information about the swarm visited positions is discarded, and a better exploitation of the past particles positions remains to be fully determined. In another recent work [18], a first promising step in that direction has been already made. In [18], the PSO has been hybridized with a machine learning algorithm, the support vector machine (SVM). The SVM has been trained on the dataset composed by all the visited swarm positions in order to build a predictive model which is able to learn where the feasible and the unfeasible regions are located in the search domain. The improvement in the managing information provided by the swarm positions during all the iterations allowed the algorithm to reduce the search space extension and considerably improve the PSO's performance. In future studies, another promising direction can be a hybridization with the estimation distribution algorithm (EDA) [57], which relies on building and updating a complex probability distribution model of the search space domain, and therefore, it is potentially able to give considerably much more information about the fitness landscape with respect to a simple blind sampling inside the search space.

**Author Contributions:** Conceptualization, M.M.R. and G.C.M.; methodology, M.M.R. and G.C.M.; software, M.M.R. and R.C.; validation, R.C. and A.A.; formal analysis, A.A.; investigation, M.M.R.; resources, M.M.R. and G.C.M.; data curation, R.C. and A.A.; writing—original draft preparation, M.M.R.; writing—review and editing, R.C. and A.A.; visualization, R.C. and A.A.; supervision, G.C.M.; All authors have read and agreed to the published version of the manuscript.

**Funding:** This research was supported by project MSCA-RISE-2020 Marie Skłodowska-Curie Research and Innovation Staff Exchange (RISE)—ADDOPTML (ntua.gr).

**Institutional Review Board Statement:** Not applicable.

**Informed Consent Statement:** Not applicable.

**Data Availability Statement:** The data used to support the findings of this study are available from the corresponding author upon reasonable request.

**Acknowledgments:** The authors would like to thank anonymous reviewers for their valuable comments and suggestions in revising the paper. The authors would like to thank G.C. Marano and the project ADDOPTML for funding supporting this research.

**Conflicts of Interest:** The authors declare no conflict of interest.

## Appendix A. Test Functions Constrained Problems

In the following, the statements of the selected benchmark numerical problems, taken by [53], which were tested in the present work are exposed.

1. **Problem g01**

   Minimize:

$$f(\boldsymbol{x}) = 5 \sum_{i=1}^{4} x_i - 5 \sum_{i=1}^{4} x_i^2 - \sum_{i=5}^{13} x_i$$

Subject to:

$$g_1(x) = 2x_1 + 2x_2 + x_{10} + x_{11} - 10 \leq 0$$
$$g_2(x) = 2x_1 + 2x_3 + x_{10} + x_{12} - 10 \leq 0$$
$$g_3(x) = 2x_2 + 2x_3 + x_{11} + x_{12} - 10 \leq 0$$
$$g_4(x) = -8x_1 + x_{10} \leq 0$$
$$g_5(x) = -8x_2 + x_{11} \leq 0$$
$$g_6(x) = -8x_3 + x_{12} \leq 0$$
$$g_7(x) = -2x_4 - x_5 + x_{10} \leq 0$$
$$g_8(x) = -2x_6 - x_7 + x_{11} \leq 0$$
$$g_9(x) = -2x_8 - x_9 + x_{12} \leq 0$$

where the search space is defined as $0 \leq x_i \leq 1$ ($i = 1, \ldots, 9$), $0 \leq x_i \leq 100$ ($i = 10, 11, 12$), $0 \leq x_{13} \leq 1$. The optimum is located at $x^* = [1; 1; 1; 1; 1; 1; 1; 1; 1; 3; 3; 3; 1]$, where $f(x) = -15$.

2. **Problem g02**

Maximize:

$$f(x) = \left| \frac{\sum_{i=4}^{n} \cos^4(x_i) - 2\prod_{i=1}^{n} \cos^2(x_i)}{\sqrt{\sum_{i=1}^{n} ix_i^2}} \right|$$

Subject to:

$$g_1(x) = 0.75 - \prod_{i=1}^{n} x_i \leq 0$$

$$g_2(x) = \sum_{i=1}^{n} x_i - 7.5n \leq 0$$

where $n = 20$ and the search space is defined as $0 \leq x_i \leq 10$ ($i = 1, \ldots, n$). The optimum OF is $f(x) = 0.803619$.

3. **Problem g04**

Minimize:

$$f(x) = 5.3578547x_3^2 + 0.8356891x_1x_5 + 37.293239x_1 - 40792.141$$

Subject to:

$$g_1(x) = 85.334407 + 0.0056858x_2x_5 + 0.0006262x_1x_4 + 0.0022053x_3x_6 \leq 92,$$
$$g_2(x) = -85.334407 - 0.0056858x_2x_5 - 0.0006262x_1x_4 + 0.0022053x_3x_6 \leq 0,$$
$$g_3(x) = 80.51249 + 0.0071317x_2x_5 + 0.0029955x_1x_2 + 0.0021813x_3^2 - 110 \leq 0,$$
$$g_4(x) = -80.51249 - 0.0071317x_2x_5 - 0.0029955x_1x_2 - 0.0021813x_3^2 + 90 \leq 0,$$
$$g_5(x) = 9.300961 + 0.0047026x_3x_5 + 0.0012547x_1x_3 + 0.0019085x_3x_4 - 25 \leq 0,$$
$$g_6(x) = -9.300961 - 0.0047026x_3x_5 - 0.0012547x_1x_3 - 0.0019085x_3x_4 + 20 \leq 0,$$

where the search space is defined as $78 \leq x_1 \leq 102$ and $33 \leq x_2 \leq 45$ and $27 \leq x_3, x_4, x_5 \leq 45$. The optimum is located at $x^* = [78, 33, 29.995256025682, 45, 36.775812905788]$, where $f(x) = -30,665.539$.

4. **Problem g06**

Minimize:

$$f(x) = (x_1 - 10)^3 + (x_2 - 20)^3$$

Subject to:

$$g_1(x) = -(x_1 - 5)^2 - (x_2 - 5)^2 + 100 \leq 0$$
$$g_2(x) = (x_1 - 6)^2 - (x_2 - 5)^2 - 82.81 \leq 0$$

where the search space is defined as $13 \leq x_1 \leq 100$ and $0 \leq x_2 \leq 100$. The optimum is located at $x^* = [14.095; 0.84296]$, where $f(x^*) = -6961.81388$.

5. **Problem g07**

Minimize:

$$f(x) = x_1^2 + x_2^2 + x_1 x_2 - 14x_1 - 16x_2 + (x_3 - 10)^2 + 4(x_4 - 5)^2 + (x_5 - 3)^2$$
$$+ 2(x_6 - 1)^2 + 5x_7^2 + 7(x_8 - 11)^2 + 2(x_9 - 10)^2 + (x_{10} - 7)^2 + 45$$

Subject to:

$$g_1(x) = -105 + 4x_1 + 5x_2 - 3x_7 + 9x_8 \leq 0$$
$$g_2(x) = 10x_1 - 8x_2 - 17x_7 + 2x_8 \leq 0$$
$$g_3(x) = -8x_1 + 2x_2 + 5x_9 - 2x_{10} - 12 \leq 0$$
$$g_4(x) = 3(x_1 - 2)^2 + 4(x_2 - 3)^2 + 2x_3^2 - 7x_4 - 120 \leq 0$$
$$g_5(x) = 5x_1^2 + 8x_2 + (x_3 - 6)^2 - 2x_4 - 40 \leq 0$$
$$g_6(x) = x_1^2 + 2(x_2 - 2)^2 - 2x_1 x_2 + 14x_5 - 6x_6 \leq 0$$
$$g_7(x) = 0.5(x_1 - 8)^2 + 2(x_2 - 4)^2 + 3x_5^2 - x_6 - 30 \leq 0$$
$$g_8(x) = -3x_1 + 6x_2 + 12(x_9 - 8)^2 - 7x_{10} \leq 0$$

where the search space is defined as $-10 \leq x_i \leq 10$ $(i = 1, \dots, 10)$. The optimum OF is $f(x^*) = 24.3062091$.

6. **Problem g08**

Maximize:

$$f(x) = \frac{\sin^3(2\pi x_1) \sin 2\pi x_2}{x_1^3(x_1 + x_2)}$$

Subject to:

$$g_1(x) = x_1^2 - x_2 + 1 \leq 0$$
$$g_2(x) = 1 - x_1 + (x_2 - 4)^2 \leq 0$$

where the search space is defined as $0 \leq x_1, x_2 \leq 10$. The optimum is located at $x^* = [1.2279713; 4.2453733]$, where $f(x^*) = -0.0958250414$.

7. **Problem g09**

Minimize:

$$f(x) = (x_1 - 10)^2 + 5(x_2 - 12)^2 + x_3^4 + 3(x_4 - 11)^2$$
$$+ 10x_5^6 + 7x_6^2 + x_7^4 - 4x_6 x_7 - 10x_6 - 8x_7$$

Subject to:

$$g_1(x) = -127 + 2x_1^2 + 3x_2^4 + + x_3 + 4x_4^2 + 5x_5 \leq 0$$
$$g_2(x) = -282 + 7x_1 + 3x_2 + 10x_3^2 + x_4 - x_5 \leq 0$$
$$g_3(x) = -196 + 23x_1 + x_2^2 + 6x_6^2 - 8x_7 \leq 0$$
$$g_4(x) = 4x_1^2 + x_2^2 - 3x_1 x_2 + 2x_3^2 + 5x_6 - 11x_7 \leq 0$$

where the search space is defined as $-10 \leq x_i \leq 10$ $(i = 1, \ldots, 7)$. The optimum OF is $f(x^*) = 680.6300573$.

8.  **Problem g12**

Maximize:

$$f(x) = \frac{100 - (x_1 - 5)^2 - (x_2 - 5)^2 - (x_3 - 5)^2}{100}$$

Subject to:

$$g(x) = (x_1 - p)^2 + (x_2 - q)^2 + (x_3 - r)^2 - 0.0625 \leq 0$$

where the search space is defined as $0 \leq x_i \leq 10$ $(i = 1, 2, 3)$ and $p, q, r = 1, 2, \ldots, 7$. The optimum OF is $f(x^*) = -1$.

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
