# Peer review of "Enhanced Multi-Strategy Particle Swarm Optimization for Constrained Problems with an Evolutionary-Strategies-Based Unfeasible Local Search Operator"

_applsci, doi:10.3390/app12052285_

Round 1
Reviewer 1 Report
Valuable work on contemporary scientific problems.Autors present study, a variant of the well-known swarm-based algorithm, the Particle Swarm Optimization (PSO).
They developed in order to solve constrained problems with a different approach with respect to the classical penalty function technique.
There is no explanation as to how the value in Equation 15 was taken.
There are some grammar errors to be corrected, as shown in the attached file.
It is recommended to carefully review the article again in order to eliminate the possible ones.

Author Response
Valuable work on contemporary scientific problems. Authors present study, a variant of the well-known swarm-based algorithm, the Particle Swarm Optimization (PSO). They developed in order to solve constrained problems with a different approach with respect to the classical penalty function technique. There is no explanation as to how the value in Equation 15 was taken.
The authors are grateful for this comment and provided a revised version of equation 15 removing it and better explaining it in the paragraph.
There are some grammar errors to be corrected, as shown in the attached file. It is recommended to carefully review the article again in order to eliminate the possible ones.
The authors are grateful for this comment and provided a revised version of the above-mentioned parts.
Reviewer 2 Report
REVIEWER’S COMMENTS
Ms. Ref. No.: applsci-1575516 (Applied Sciences)
Manuscript Title: Enhanced multi-strategy Particle Swarm Optimization for constrained problems with an Evolutionary-Strategies-based unfeasible local search operator
Author/Authors: Marco Martino Rosso, Raffaele Cucuzza, Angelo Aloisio and Giuseppe Carlo Marano.
In this study, a variant of Particle Swarm Optimization (PSO) is implemented for the purpose of studying a different way to deal with constrained optimization problems and the PSO is hybridized with a machine learning algorithm, the support vector machine (SVM) by the author.
The problem itself, in this reviewer’s opinion, is not of particular significance. Hence, there is no valuable finding in the paper. Accordingly, I do not feel that this paper has enough new and important materials to constitute an archival full-length paper.
It seems that the manuscript has been prepared in rush; most parts of the manuscript are poorly written and need many corrections/modifications. Moreover, there are numerous writing errors which create confusion for the reader. English of the paper have to be improved throughout the text to make it consistent with the journal standard. I don't think this paper is suitable for publication in Applied Sciences.

Author Response
In this study, a variant of Particle Swarm Optimization (PSO) is implemented for the purpose of studying a different way to deal with constrained optimization problems and the PSO is hybridized with a machine learning algorithm, the support vector machine (SVM) by the author.
The authors are grateful for this comment, but unfortunately, reviewer two's remarks seem to be referred to topics that are out of the present paper's scope and are not treated in the current research. Moreover, the reviewer appears to express this comment only based on the conclusion part, without considering the entire manuscript. Indeed, the present study refers to improving the basic PSO implementation with a multistrategy approach. Besides, the novel hybridization refers to integrating another meta-heuristic algorithm called Evolutionary Strategies (ES) to perform an unfeasible local search operator. Therefore, the reviewer's comment about integrating the machine learning algorithm, i.e. the support vector machine, appears to be entirely inappropriate in the current research. However, the authors are grateful to the reviewer for his frank comments. To avoid further misunderstanding, the conclusions have been revised.
The problem itself, in this reviewer’s opinion, is not of particular significance. Hence, there is no valuable finding in the paper. Accordingly, I do not feel that this paper has enough new and important materials to constitute an archival full-length paper.
The authors are grateful for this comment. To point out the finding of the present implementation, some real-world structural optimization case studies have been added as validation of the proposed framework. Additionally, the authors compared the PSO-multistrategy results with PSO with a static and dynamic penalty and a Genetic Algorithm (GA).
It seems that the manuscript has been prepared in rush; most parts of the manuscript are poorly written and need many corrections/modifications. Moreover, there are numerous writing errors which create confusion for the reader. English of the paper have to be improved throughout the text to make it consistent with the journal standard. I don't think this paper is suitable for publication in Applied Sciences.
The authors are grateful for this comment. We revised the grammatical errors pointed out even by other reviewers. However, as declared by the other reviewers, the English of the current paper does not appear out of order with the current Applied Sciences standards (English language and style are fine/minor spell check required). In any case, the authors are grateful for this comment and attempted to do their best to revise the entire manuscript to make it clear fluent and help the reader fully understand the research findings by reporting some bullet points summary.
Reviewer 3 Report
- The manuscript is concerned with the enhanced multi-strategy Particle swarm optimization for constrained problems with an unfeasible local search. It is relevant and within the scope of the journal.
- However, the manuscript, in its present form, contains several weaknesses. Adequate revisions to the following points should be undertaken in order to justify recommendations for publication.
- For readers to quickly catch the contribution in this work, it would be better to highlight major difficulties and challenges, and your original achievements to overcome them, in a clearer way in the abstract and introduction.
- It is shown in the reference list that the authors have several publications in this field. This raises some concerns regarding the potential overlap with their previous works. The authors should explicitly state the novel contribution of this work, the similarities, and the differences of this work with their previous publications.
- A variant of the well-known PSO algorithm, the Particle Swarm Optimization (PSO), is developed in order to solve constrained problems with a different approach with respect to the classical penalty function technique. What are the other feasible alternatives? What are the advantages of adopting these soft computing techniques over others in this case? How will this affect the results? More details should be furnished.
- The combination of PSO and local search mechanism with feasible and non-feasible solution utilization is adopted for constraint optimization problems. What are the other feasible alternatives? What are the advantages of adopting this approach over others in this case? How will this affect the results? More details should be furnished.
- The authors have used the penalty function approach for constraint handling. What are the other feasible alternatives? What are the advantages of adopting these solutions over others in this case? How will this affect the results? More details should be furnished.
- What are the stochastic parameter settings adopted for the compared algorithms? What are the other feasible alternatives? What are the advantages of adopting these settings over others in this case? How will this affect the results? More details should be furnished.
- Only three versions of PSO are adopted as benchmarks for comparison. What are the other feasible alternatives? What are the advantages of adopting these methods over others in this case? How will this affect the results? More details should be furnished.
- The comparison is very poor, further comparisons are required with state-of-the-art algorithms such as grey wolf optimizer, multi-verse optimizer, whale optimization algorithm, Harris Hawk algorithm, gradient-based optimizer, stochastic fractal search, etc.
- A non-parametric statistical test is required when comparing with the above-mentioned algorithms.
- The 13 numerical constraint optimization functions used for testing are insignificant. Further tests MUST be studied such as the real-world constraint optimization test suites of congress of evolutionary computation (CEC).
- Some assumptions are stated in various sections. More justifications should be provided on these assumptions. Evaluation on how they will affect the results should be made.
- The discussion in the present form is relatively weak and should be strengthened with more details and justifications. A separate discussion section should be added to the manuscript.
- The contributions and novelty of the paper should be added before the last paragraph in the introduction section.
- The organization of the rest of the paper should be written as the last paragraph in the introduction section.
- The future scope should be added the conclusion section.
Author Response
The manuscript is concerned with the enhanced multi-strategy Particle swarm optimization for constrained problems with an unfeasible local search. It is relevant and within the scope of the journal. However, the manuscript, in its present form, contains several weaknesses. Adequate revisions to the following points should be undertaken in order to justify recommendations for publication. For readers to quickly catch the contribution in this work, it would be better to highlight major difficulties and challenges, and your original achievements to overcome them, in a clearer way in the abstract and introduction.
The authors are grateful for this comment. A global revision has been provided to improve the quality of the abstract. Furthermore, bullet points have been added to the introduction to help the reader fully understand the objectives and novelties of this research.
It is shown in the reference list that the authors have several publications in this field. This raises some concerns regarding the potential overlap with their previous works. The authors should explicitly state the novel contribution of this work, the similarities, and the differences of this work with their previous publications.
The authors are grateful for this comment. They declare that there is no overlap with other authors' previous work. Every result presented here is original and obtained from Matlab code implementation. Furthermore, the PSO variant shown here is different from those discussed in previous works. Even the structural optimization benchmark provided here is entirely different from other authors' previous work. However, the authors are very grateful for the reviewer comment. To help the reader fully understand the novelty and achievements of the present research, some bullet points summary have been added in the text. The authors added the following sentences in the introduction, expressing the novelty of this contribution compared to other papers by the authors: “In a different recent contribution of the authors [55], some further novel approaches to deal with constraints have been presented, considering an hybridization of PSO with machine learning support vector machine. However, the current paper presents a completely different approach based on constraints handling directly based on information which can be retrieved from the swarm positions, in terms of objective function and constraints violations.”
A variant of the well-known PSO algorithm, the Particle Swarm Optimization (PSO), is developed in order to solve constrained problems with a different approach with respect to the classical penalty function technique. What are the other feasible alternatives? What are the advantages of adopting these soft computing techniques over others in this case? How will this affect the results? More details should be furnished.
The authors are grateful for this comment. The main advantage provided by the soft computing techniques is related to the fact that the meta-heuristic algorithm does not require a complex mathematical formulation of the problem because they attempt to solve the problem by combining stochastic search procedures by mimicking natural phenomena. Moreover, since the fact mentioned above, they do not require information of the gradient or the Hessian of the objective function, which can be very computationally hard to solve, especially with highly non-linear problems such as those involved by structural optimization tasks. The state-of-art of the main constraint handling approaches have already been reported in Section 2.1, highlighting the limits and arbitrary of some nowadays adopted techniques. Therefore, the presented framework relies directly on the actual evaluation of the objective function and the constraints violations, reducing the arbitrary of the constraint handling approach, which effectively exploits information that has been already retrieved by the entire swarm positions in the search space.
The combination of PSO and local search mechanism with feasible and non-feasible solution utilization is adopted for constraint optimization problems. What are the other feasible alternatives? What are the advantages of adopting this approach over others in this case? How will this affect the results? More details should be furnished.
The authors are grateful for this comment. In the presented research, the PSO evolution towards the best solution relies directly on evaluating the objective function and the constraints violations. Thus, it reduces the arbitrary constraint handling approach, which effectively exploits information already retrieved by the entire swarm positions in the search space. The authors added the following sentences in the introduction, expressing the novelty of this contribution compared to other constraints handling approaches “In a different recent contribution of the authors [55], some further novel approaches to deal with constraints have been presented, considering a hybridization of PSO with machine learning support vector machine. However, the current paper presents a completely different approach based on constraints handling directly based on information which can be retrieved from the swarm positions, in terms of the objective function and constraints violations.”
The authors have used the penalty function approach for constraint handling. What are the other feasible alternatives? What are the advantages of adopting these solutions over others in this case? How will this affect the results? More details should be furnished.
The authors are grateful for this comment. The penalty function has been adopted only for completeness to make a relevant comparison with the proposed method and the nowadays most widespread technique. The authors criticize that the most widespread method is due to its simplicity of implementation only, but not always it is a good choice, especially for some highly non-linear structural optimization problems. Therefore, to the authors knowledge, comparing the PSO-multistrategy and some widespread PSO implementation represents an good way to show that even without some arbitrary penalty approach, the PSO-multistrategy can find the optimal solution of the same order of magnitude and, sometimes, even with better results.
What are the stochastic parameter settings adopted for the compared algorithms? What are the other feasible alternatives? What are the advantages of adopting these settings over others in this case? How will this affect the results? More details should be furnished.
The authors are grateful for this comment. Some PSO with penalty approaches has been implemented to perform some relevant comparison with PSO-multistrategy. Their stochastic parameter choices have been always reported in the text, or at least the references of the implementations have been provided.
Only three versions of PSO are adopted as benchmarks for comparison. What are the other feasible alternatives? What are the advantages of adopting these methods over others in this case? How will this affect the results? More details should be furnished.
The authors are grateful for this comment. Some further real-world structural optimization problems have been added to the present research to improve the quality of the study and underline the capabilities of the proposed PSO implementation to deal with highly non-linear combinatorial engineering problems. In these final comparisons, the authors perform comparisons even with Genetic Algorithm Matlab implementation for completeness.
The comparison is very poor, further comparisons are required with state-of-the-art algorithms such as grey wolf optimizer, multi-verse optimizer, whale optimization algorithm, Harris Hawk algorithm, gradient-based optimizer, stochastic fractal search, etc.
The authors are grateful for this comment. Multiple real-world structural optimization problems have been added to the present research to improve the quality of the study and underline the capabilities of the proposed PSO implementation to deal with highly non-linear combinatorial engineering problems. In these final comparisons, the authors perform comparisons even with Genetic Algorithm Matlab implementation for completeness.
A non-parametric statistical test is required when comparing with the above-mentioned algorithms. The 13 numerical constraint optimization functions used for testing are insignificant. Further tests MUST be studied such as the real-world constraint optimization test suites of congress of evolutionary computation (CEC).
The authors are grateful for this comment. Since the intrinsic nature of meta-heuristic, considering standard literature approaches, the comparison has been performed based on a great number of independent runs of the algorithm and results have been critically analyzed in terms of accuracy of the final solution and overall variability in terms of the standard deviation of the objective function results. Standard deviation provide already a statistical feature to make a comparison among all meta-heuristic algorithms results. Moreover, in order to provide some more consistent results, some further state-of-art real-world structural optimization problems have been added to the present research in order to improve the quality of the research and underline the capabilities of the proposed PSO implementation to deal with highly non-linear combinatorial engineering problems. In these final comparisons, for the sake of completeness, the authors perform comparisons even with Genetic Algorithm Matlab implementation.
Some assumptions are stated in various sections. More justifications should be provided on these assumptions. Evaluation on how they will affect the results should be made.
The authors are grateful for this comment. The authors attempted to do their best by revising the entire manuscript and giving all the necessary references or discussion when some parameters have been assumed.
The discussion in the present form is relatively weak and should be strengthened with more details and justifications. A separate discussion section should be added to the manuscript.
The authors are grateful for this comment. The authors provided a final brief discussion section in which the PSO-multistrategy results have been critically discussed in comparison with numerical benchmark and final real-world highly non-linear structural optimization engineering problems.
The contributions and novelty of the paper should be added before the last paragraph in the introduction section.
The authors are grateful for this comment. In order to help the reader fully understand the novelty and achievements of the present research, some bullet points summary have been added in the text following the reviewer indications.
The organization of the rest of the paper should be written as the last paragraph in the introduction section.
The authors are grateful for this comment. The author provided a revised version of the last paragraph of the introduction section.
The future scope should be added the conclusion section.
The authors are grateful for this comment. The authors pointed out future developments in the conclusion section.
Round 2
Reviewer 3 Report
The authors have replied to all my comments and made the necessary changes. Hence, the paper can be accepted for publication.